# IS CLASSIFICATION ALL YOU NEED FOR RADIOLOGY REPORT GENERATION?

## ABSTRACT

Automatic radiology report generation is an advanced medical assistive technology capable of producing coherent reports based on medical images, akin to a radiologist. However, current generative methods exhibit a notable gap in clinical metrics when compared to medical image classification. Recently, leveraging diagnostic results to improve report quality has emerged as a promising approach. We are curious whether training a classifier that encompasses all possible long-tailed and rare diseases could enhance the robustness of reports. To investigate this question, this study designs an evaluation framework that integrates long-tail scenarios and summarizes potential combinations of LLM-based report generation models. We assess the impact of classification on report quality across four benchmarks. Initially, we introduce LLM-based language and clinical metrics and develop a pipeline to evaluate the model's performance on both in-domain and out-of-distribution (OOD) long-tail scenarios. Subsequently, we conduct a systematic evaluation of all potential model combinations. Our findings reveal that: 1) the impact of classification on report quality is positively correlated with the performance of classifiers, but the gap still exists, and 2) while classification can enhance report quality in in-domain long-tail scenarios, its benefits for OOD scenarios are limited.

## 1 INTRODUCTION

Automatic radiology report generation (ARRG) (Jing et al., 2017) has emerged as a significant research area within medical imaging and natural language processing (NLP). The objective of ARRG systems is to accurately generate comprehensive and clinically meaningful reports from medical images, which has the potential to alleviate the workload of radiologists, reduce diagnostic errors, and improve patient outcomes. Furthermore, such systems can enhance accessibility to high-quality healthcare by providing diagnostic support in regions with limited medical resources.

Despite significant advances in deep learning for medical image analysis, generating coherent and precise medical reports remains highly challenging due to the complexity of visual information and the nuances of medical language. Traditional methods, such as retrieval-based (Li et al., 2019; 2018) and template-based (Biswal et al., 2020; Harzig et al., 2019; Li et al., 2018) approaches, often rely on fixed rules or knowledge closely tied to training data for generating radiology reports. In recent years, LLM-based methods (Li et al., 2024; Bannur et al., 2024; Tu et al., 2024) have become an attractive research direction, leveraging the powerful extrapolation and in-context learning capabilities to enhance the accuracy of generated reports and improve the interactivity of ARRG systems. However, when evaluating the diagnostic accuracy on specific radiology findings, we observe that the accuracy of reports generated by existing methods falls significantly short compared to the performance of basic medical image classification approaches. For example, on the MIMIC-CXR dataset, state-of-the-art generation models exhibit accuracy that is at least 20% lower than that of image classification models[1]. To address this issue, some studies (Wang et al., 2023; Zhao et al., 2024b; Jin et al., 2024) have sought to integrate diagnostic results to improve the accuracy and reliability of generated reports. This raises an important question: **is classification all you need for radiology report generation?** Specifically, this paper aim to conduct a comprehensive study to determine whether training a classifier that encompasses all possible radiology findings in the training data,

---

[1]The details are shown in the appendix.

including long-tailed and rare diseases, would improve the robustness of report generation when its diagnostic results are incorporated.

To validate this hypothesis, we design a benchmark framework by modifying the existing evaluation setup and introducing a set of baseline methods from a newly proposed LLM based design space for report generation. Current evaluation metrics primarily include language metrics and clinical metrics. Language metrics, such as BLEU (Papineni et al., 2002) and ROUGE (Lin, 2004), focus on n-gram overlap and sequence alignment, while clinical metrics, like CheXpert F1 (Smit et al., 2020) and RadGraph metric (Jain et al., 2021), emphasize clinical events described in radiology reports, such as pathological entities, their locations, and severity, based on predefined categories. Conventional language metrics, however, primarily focused on grammatical and lexical similarities, often fail to accurately reflect the precision required in clinical diagnostic reports. Clinical metrics, constrained by a limited set of predefined categories, struggle to capture the intricate diversity of clinical scenarios depicted in medical documents. Furthermore, current clinical metrics are not well-equipped to evaluate nuanced distinctions in inclusive relationships (e.g., differentiating between the left upper lobe and the entire left lung) and near synonyms (e.g., distinguishing a nodule from an opacity). To address these challenges, besides these conventional evaluation metrics, we proposed to introduces two extra metrics that leverages large language models (LLM) to mitigate the shortcomings of both metric types. LLM-based language metrics provide analytical capabilities that transcend simple sentence similarity, enabling the comprehension of clinical terminology for assessing report similarity. We reference LLM-RadJudge (Wang et al., 2024) as our LLM-based language metric, an LLM-based language metric that evaluates report similarity across six distinct levels. Additionally, we propose a clinical metric based on LLM that automatically extracts all possible radiology findings from reports, including long-tail and rare disease categories that may not be part of any predefined finding set.

In comparing baselines in our study, we propose an LLM-based design space for report generation models, outlining three key components of existing LLM-based methods: the vision encoder, classifiers, and the LLM itself. The vision encoder, such as CLIP and DINO, extracts abstract features from medical images, transforming them into vision tokens. The classifier derives easily interpretable radiology observations, including probabilities and diagnosis confidence, which can be represented as classification tokens for further processing. The LLM module aggregates and processes all tokens, generating reports in an auto-regressive manner. We analyzed potential combinations of these components and identified four baseline models, as illustrated in Fig. 1, which align with the model design space of most existing methods.

Under the proposed benchmark framework, we conducted extensive experiments across four benchmarks, revealing a counterintuitive phenomenon: while diagnostic results significantly improve report quality in in-domain scenarios, they do not enhance report quality for long-tail diseases and out-of-distribution (OOD) data scenarios when using powerful foundation models such as Llama 3.1 70B (Dubey et al., 2024) and OpenAI GPT-4. To better understand this phenomenon, we performed detailed case studies and analyses. We found that LLM effectively utilize information provided by classifiers to generate final reports, sometimes including observations not mentioned in actual clinical reports. Consequently, the information from the classifier may mislead LLM-based generative methods, leading to incorrect results. In summary, our findings indicate that **incorporating additional classification information can enhance report quality in in-domain scenarios but may severely compromise performance in zero-shot settings.** Experimental results demonstrate that LLM can modify and augment original reports based on classification information, potentially correcting initial erroneous conclusions and narrowing the gap between classification and generation. **At the same time, it will also amplify the misclassification errors of the classifier in the long-tail scenario.** We hope these findings will inspire further exploration of LLM-based metrics and classification-based report generation in this field.

## 2 METHODS

We aim to explore where classifier-based methods help and why. In this section, we introduce how to explore the model design space and robustness evaluation framework to understand the gap between report generation and classification.

## 2.1 BASIC SETUP

For convenience, we employ LLaVA's (Liu et al., 2024a) model architecture as the basis, which consists of a large language model, a vision encoder, and a connector. The connector projects the visual embedding from the vision encoder into the text embedding space. The connector is a multi-layer perceptron (MLP) with GELU activations (Huang et al., 2023) and a hidden size of 1024 for all layers.

## 2.2 MODEL DESIGN SPACE

Our goal is to explore potential model's architecture from a high-level perspective, as illustrated in Fig. 1. The existing LLM-based generation methods primarily consist of three main components: a vision encoder, a classifier, and an LLM.

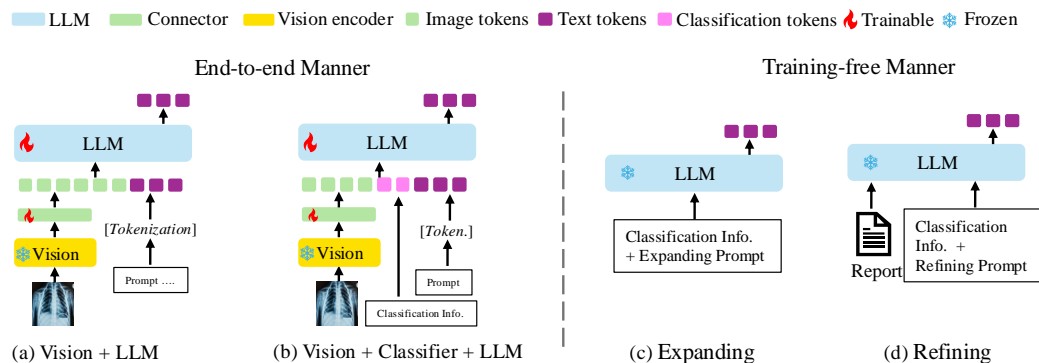

Figure 1: We combined the vision encoder, classifier, and large language model (LLM) from two training perspectives, resulting in five potential LLM-based report generation methods represented abstractly.

**The vision encoder** is a sophisticated component designed to meticulously analyze and extract abstract features from medical images. The extracted features are then meticulously transformed into *vision tokens*, which are essentially compact representations of the original data that retain the essential information for further analysis.

**The classifier** plays a pivotal role in the system by deriving easily interpretable observational information from the vision tokens. It calculates probabilities and confidence levels for various potential diagnoses, which are critical for medical decision-making. These statistical measures are then converted into *classification tokens*.

**The LLM (Large Language Model)** module serves as the central hub of the system, where it aggregates and processes all the tokens generated by the vision encoder and classifier. It leverages its extensive training on vast amounts of medical literature and data to generate comprehensive and coherent reports. These reports are crafted in an auto-regressive manner, ensuring that each subsequent part of the report is informed by the context established by the previous sections.

Based on the characteristics of the training paradigms, we can categorize the existing methods into end-to-end training and training-free approaches. Among all training methods, the LLM is an essential module, allowing us to focus on the combination of the vision encoder and the classifier.

**The classifier as input to LLM.** It employs only the classifier, with the large language model (LLM) building upon the classification information to generate reports. This approach yields the paradigms illustrated in Fig. 1c and 1d. Additionally, as shown in Fig. 1d, we can allow the LLM to refine a report using the classification information, enhancing the accuracy of the generated report. Researches (Wang et al., 2023; Zhao et al., 2024b) belong to this paradigm.

**The vision encoder as input to LLM.** Conversely, as illustrated in Fig. 1a, using only the vision encoder, with the large language model (LLM) building upon the vision encoder to generate reports, as seen in (Hyland et al., 2023; Dubey et al., 2024).

**Hybrid input to LLM.** It is to combine these three modules to obtain the paradigm shown in Fig. 1b, which is utilized in (Jin et al., 2024). In practical applications, selecting different backbones for each module can lead to significantly varied results; we discuss common backbones in the appendix. In our experiments, we default to using Rad-DINO (Pérez-García et al., 2024) as the vision encoder and Swin Transformer-Large (Taslimi et al., 2022) as the classifier.

## 2.3 EVALUATION FRAMEWORK

In this part, our objective is to design a comprehensive benchmark framework that can evaluate LLM-based methods in long-tail scenarios. We assess the robustness of a report generation model from both in-domain and out-of-distribution (OOD) perspectives. First, we introduce the training datasets, as well as the long-tail datasets for both in-domain and OOD scenarios. Next, we present our evaluation metrics, which include traditional language and clinical metrics, alongside LLM-based metrics.

### 2.3.1 DATASETS AND DATA PRE-PROCESSING

We train all baselines on MIMIC-CXR 2.0.0 (Johnson et al., 2019c;b), a large dataset of chest radiographs in DICOM format with free-text radiology reports, containing 377,110 images corresponding to 227,835 radiographic studies. Following (Hyland et al., 2023), we extract the *Findings* and *Indication* sections for each report, and discard all studies for which *Findings* could not be extracted. Unlike (Hyland et al., 2023), we used png files from MIMIC-CXR-JPG (Johnson et al., 2019a) as vision inputs, instead of the original DICOM files, the former of which show better compatibility in our experimental setup. Following (Hyland et al., 2023), we used the available finding parts from the official MIMIC-CXR test split, totaling 2,461 samples.

For evaluation, we utilized four datasets: MIMIC-CXR, CXR-LT (Holste et al., 2023; Goldberger et al., 2000), PadChest (Bustos et al., 2020), and IU X-Ray (Demner-Fushman et al., 2016). We divide them into in-domain and OOD long-tail datasets:

**The In-domain Long-tail Dataset.** We use CXR-LT as a dataset to verify the performance of the model in the in-domain longtail scenario. It is an extension version of MIMIC-CXR, to evaluate the performance of report generation models in long-tailed scenarios, containing 377,110 CXRs from 26 long-tail observations.

**The OOD Long-tail Dataset.** We introduce two out-of-distribution (OOD) long-tail datasets to evaluate the model's generalization capability in different X-ray positions as well as different languages and reporting styles. PadChest is a large-scale, high-resolution, labeled chest X-ray dataset designed for automated medical image analysis, accompanied by corresponding reports. It contains over 160,000 images from 67,000 patients. From this dataset, we randomly sampled 500 instances, comprising 99 observations, to form the test set. Additionally, we employed GPT-4 to translate the original Spanish reports into English. The IU X-Ray dataset consists of 7,470 chest X-ray images paired with diagnostic reports. We categorized the labels based on primary disease classifications, yielding a test set of 756 samples across 82 observations.

For image processing, we resize all images to $224 \times 224$ and $518 \times 518$ to adapt various vision backbones, e.g., CLIP (Radford et al., 2021) and Rad-DINO (Pérez-García et al., 2024), and we do not apply any data augmentation to images. For text processing, we utilize a same processing pipeline in (Chiang et al., 2023).

### 2.3.2 EVALUATION METRICS

To more comprehensively evaluate the performance of different methods, we introduce the following metrics: language, clinical, and LLM-based metrics, including:

**Language Metrics.** *ROUGE-L* (Lin, 2004), this metric assesses the length of the longest common subsequence of words, normalized by the lengths of both the predicted and reference texts; *BLEU-1/-4* (Papineni et al., 2002), these metrics evaluate n-gram precision, with BLEU-1 focusing on single words and BLEU-4 considering up to four-word sequences. A brevity penalty is applied to mitigate the impact of excessively short predictions; *METEOR* (Banerjee & Lavie, 2005), this metric aligns

unigrams from the prediction and reference texts while maintaining their order, and calculates a weighted harmonic mean of precision and recall, incorporating a penalty for fragmented sequences.

**Clinical Metrics.** *CheXpert F1*, this metric utilizes the CheXbert automatic labeler (Smit et al., 2020) to categorize observations into 'present', 'absent', or 'uncertain' for each of the 14 CheXpert pathological conditions. We provide macro- and micro-averaged F1 scores for both the 5 major observations and all 14 observations, termed "[Macro/Micro]-F1-[5/14]"; *RadGraph metric* (Jain et al., 2021; Delbrouck et al., 2022), this metric measures the overlap of entities and relations separately, and then computes their average. Entities are matched based on identical text spans and types, while relations are matched based on the endpoints and the relation type, termed $RG_{ER}$ score. This evaluation is conducted using the radgraph package[2].

**LLM-based Metrics.** *LLM-Radjudge* (Wang et al., 2024) uses large language models to assess the quality of radiology reports, providing a detailed description and classification of errors. It includes six error levels: levels 1 and 2 describe the number of observational errors, level 3 describes the number of location errors, level 4 describes the number of severity errors, and levels 5 and 6 compare with previous reports. We report the average values for levels 1-4; *Long-tailed & OOD F1*, this metric is used to validate the generalization ability of the generation model to diseases that have never been seen in the training set. We use the OpenAI GPT-4o API to extract the disease categories from the generated reports and then compute the F1 score of these extracted categories. Specifically, we use "[Macro/Micro]-F1-[LT26/LT99/OOD82]" to represent the result on 26 observations in the CXR-LT dataset, 99 observations in the PadChest dataset, and 82 observations in the IU X-ray dataset, respectively. **Note that** the metrics used in this study indicate that higher values are better for all metrics except LLM-Radjudge where lower values are better.

## 3 EXPERIMENTS

In this section, we study the impact of various component variants on the quality of report generation. Specifically, we focus on the components of vision encoders, the classifier, and LLM. In our setting, we examine four different vision encoders, namely Swin Transformers (Liu et al., 2021), Rad-DINO (Pérez-García et al., 2024), ViT-L (Dosovitskiy et al., 2021), and DINOv2 (Oquab et al., 2024). For the LLM, we use off-the-shelf models of different scales, such as Phi3-3B (Abdin et al., 2024), Vicuna-1.5-7B/13B (Zheng et al., 2024). The implementation details are shown in Appendix.

### 3.1 THE ROLE OF CLASSIFIER IN IN-DOMAIN LONG-TAIL SCENARIO

**Does the classifier help in conventional report generation?** We compared our approach with four state-of-the-art (SOTA) baselines on the MIMIC-CXR dataset using traditional evaluation metrics. In our settings, we used Swin Transformer-Large as the classifier, Rad-DINO as the vision encoder, and Vicuna-1.5 (7B) as the large language model (LLM). We used four SOTA methods for comparison: RGRG (Tanida et al., 2023), R2GEN (Chen et al., 2020), MAIRA-1 (Hyland et al., 2023), and ChatCAD+ (Zhao et al., 2024b). As shown in Table 1, the results indicate that directly 'Expanding' classification information using an LLM yields the lowest performance among the four baselines. In contrast, the 'Refining' method demonstrates superior performance across most metrics. We believe that expanding without any additional information makes it difficult to produce a reliable report. Moreover, the comparison between 'V+LLM' and 'C+V+LLM' shows that incorporating classification information effectively improves the classification performance of the report. Finally, the comparison between 'C+V+LLM' and 'Refining' reveals that while 'Refining' achieves the most significant improvements in in-domain scenarios across most metrics, its performance in long-tail scenarios is inferior to the end-to-end training paradigm of 'C+V+LLM'.

**Does the classifier help on in-domain's long-tail scenarios of report generation?** To equip the classifier based method with recognition ablity on long-tail data, we initially trained classifiers utilizing long-tail categories extracted from the MIMIC-CXR dataset. For simplicity, we employed the Swin Transformer-Large to train on 100 and 200 long-tail categories, respectively. Subsequently, we conducted experiments that combined the classification results with various scales of LLM. Additionally, from the perspective of the model architecture, we categorized these baselines into trainable and frozen weights, with results presented in Table 2. A substantial number of experimental results

---

[2]https://pypi.org/project/radgraph/

Table 1: We report the performance of various models on the MIMIC-CXR dataset. 🔥 and ❄️ indicates whether the backbone is trainable or frozen, respectively. 'C' represents the classifier, 'V' is the vision encoder. The **bold** indicates the best value. † indicates that the result is directed cited from the original paper. *RJ-n* represents the level score of LLM-Radjudge.

| Metrics | state-of-the-art | | | | Our baselines | | | |
|---|---|---|---|---|---|---|---|---|
| | RGRG† | R2Gen† | MAIRA-1 | ChatCAD+† | 🔥V + LLM | 🔥C + V + LLM | ❄️Expanding | ❄️Refining |
| ROUGE-L | 26.4 | 27.7 | 29.8 | 17.4 | 28.9 | 29.9 | 19.8 | **30.1** |
| BLUE-1 | 37.3 | 35.3 | 37.7 | 31.6 | 37.7 | 34.8 | 27.5 | **38.5** |
| BLUE-4 | 12.6 | 10.3 | 14.2 | 0.8 | 14.6 | 13.6 | 5.5 | **16.0** |
| METEOR | 16.8 | 14.2 | 33.2 | 24.1 | 32.3 | 31.8 | 22.7 | **33.4** |
| $RG_{ER}$ | - | - | 29.0 | - | 29.0 | 28.0 | 19.2 | **29.7** |
| RJ-1 | - | - | 0.24 | - | 0.24 | 0.26 | 0.47 | **0.21** |
| RJ-2 | - | - | 2.77 | - | 2.76 | 2.79 | 2.89 | **2.53** |
| RJ-3 | - | - | **0.15** | - | **0.15** | **0.15** | 0.71 | 0.26 |
| RJ-4 | - | - | **0.09** | - | **0.09** | 0.11 | 0.24 | 0.13 |
| Macro-F1-5 | - | - | 46.1 | 47.4 | 46.1 | **48.9** | 38.6 | 46.4 |
| Micro-F1-5 | 54.7 | - | 54.8 | - | 54.8 | 55.5 | 47.7 | **55.7** |
| Macro-F1-14 | - | 27.6 | 36.7 | - | 35.7 | **38.7** | 25.5 | 38.2 |
| Micro-F1-14 | 44.7 | - | 54.6 | - | 54.6 | 52.9 | 34.2 | **55.9** |
| Macro-F1-LT26 | - | - | 21.4 | - | 21.4 | **30.9** | 8.6 | 13.8 |
| Micro-F1-LT26 | - | - | 43.1 | - | 43.1 | **45.9** | 21.7 | 29.6 |

Table 2: The impact of long-tail classifiers on different methods on the CXR-LT dataset. 🔥 and ❄️ indicates whether the backbone is trainable or frozen, respectively. *RJ-n* represents the level score of LLM-Radjudge.

| Method | ROUGE-L | BLUE-1/-4 | METEOR | $RG_{ER}$ | RJ-1 | RJ-2 | Macro-F1-14 | Macro-F1-LT26 |
|---|---|---|---|---|---|---|---|---|
| *Classifier (Swin Transformer-L)* | | | | | | | | |
| LT-100 | - | - | - | - | - | - | 57.1 | 49.1 |
| LT-200 | - | - | - | - | - | - | 56.0 | 47.3 |
| *Baseline (V: Rad-DINO)* | | | | | | | | |
| 🔥 Phi-3-3B | 29.9 | 35.3 / 14.1 | 32.3 | 27.9 | 0.28 | 2.64 | 34.7 | 20.7 |
| 🔥 Phi-3-3B + LT-100 | 29.8 | 35.4 / 13.9 | 32.1 | 27.7 | 0.28 | 2.66 | 36.5 | 29.1 |
| 🔥 Phi-3-3B + LT-200 | 29.8 | 35.0 / 13.8 | 32.1 | 27.7 | 0.27 | 2.66 | 36.5 | 27.7 |
| 🔥 Vicuna-1.5-7B | 29.8 | 37.7 / 14.6 | 33.2 | 29.0 | 0.24 | 2.77 | 36.7 | 21.4 |
| 🔥 Vicuna-1.5-7B + LT-100 | 30.1 | 36.6 / 14.5 | 32.9 | 28.3 | 0.26 | 2.79 | 36.6 | 30.9 |
| 🔥 Vicuna-1.5-7B + LT-200 | 30.0 | 36.4 / 14.4 | 32.8 | 28.3 | 0.26 | 2.79 | 36.6 | 30.1 |
| 🔥 Vicuna-1.5-13B | 30.0 | 38.1 / 14.9 | 32.2 | 27.9 | 0.24 | 2.76 | 37.9 | 20.3 |
| 🔥 Vicuna-1.5-13B + LT-100 | 30.3 | 36.4 / 14.3 | 32.8 | 28.0 | 0.26 | 2.78 | 39.0 | 31.3 |
| 🔥 Vicuna-1.5-13B + LT-200 | 30.3 | 36.4 / 14.3 | 32.8 | 28.0 | 0.26 | 2.78 | 39.0 | 31.0 |
| ❄️ Llam3.1-70B | 19.8 | 27.5 / 5.5 | 22.7 | 19.2 | 0.47 | 2.89 | 25.5 | 8.6 |
| ❄️ Llam3.1-70B + LT-100 | 19.8 | 27.3 / 5.4 | 22.7 | 19.2 | 0.44 | 2.85 | 25.6 | 13.7 |
| ❄️ Llam3.1-70B + LT-200 | 19.8 | 27.3 / 5.4 | 22.7 | 19.2 | 0.44 | 2.85 | 25.6 | 12.0 |
| ❄️ ICL + Llam3.1-70B | 19.6 | 27.1 / 5.0 | 22.6 | 19.1 | 0.49 | 2.93 | 22.8 | 8.4 |
| ❄️ ICL + Llam3.1-70B + LT-100 | 19.6 | 27.1 / 5.0 | 22.6 | 19.1 | 0.49 | 2.93 | 22.8 | 10.7 |
| ❄️ ICL + Llam3.1-70B + LT-200 | 19.6 | 27.1 / 5.0 | 22.6 | 19.1 | 0.49 | 2.93 | 22.8 | 10.0 |

demonstrate that baselines utilizing the long-tail classifier significantly enhance long-tail classification capabilities in the in-domain context compared to the baseline that do not use the classification information.

**Finding 1:** *The classifier can improve the classification performance of generated reports in both in-domain and long-tail scenarios.*

## 3.2 ABLATION STUDY ON THE DESIGNS OF THE INDIVIDUAL MODULES

**Variation of classification information representation.** Introducing classification information can be done by adding semantic-level tokens, such as additional *[CLS]* tokens, and by directly converting the classification information into a prompt as input to the model. We conducted the following experiments to answer this question. 1) Using only a single image *[CLS]* token. As shown in Fig. 2, we used variants of vision encoders, including Swin transformers-L and Rad-DINO fine-tuned on chest X-ray images, as well as ViT-L and DINOv2 pre-trained on ImageNet-21k. We fine-tuned our report generation model using their *[CLS]* tokens and patch tokens (w/ *[CLS]* token), as well as using only patch tokens (w/o *[CLS]* token); 2) Adding *[CLS]* tokens to the text and image, respectively; 3) Adding a special binary classification *[CLS]* token for each observation, indicating whether the corresponding observation is positive or negative. The results of experiments 2 and 3 are shown in Table 3a; 4) The output probability of the classifier is converted into a prompt, and the results are shown in Table 3b. The format of the prompt is "*The [obs.] is [positive / negative] (Probability: [x]%)*". Note that, all *[CLS]* tokens pass through LLM. Overall, the results suggest that adding *[CLS]* tokens to the input, even from high-performing classifiers, does not substantially improve report generation performance. This phenomenon is contrary to the conclusion of (Kim et al., 2021; Touvron et al., 2021). However, using classification information directly as a prompt input is a more effective strategy for improving the model's classification ability, which aligns with previous findings in prompt-based learning studies (Wang et al., 2023; Jin et al., 2024).

Based on previous results, we aim to seek a way that can effectively improve the generalization of generated reports, specifically by using large language models (LLM) to refine the original reports based on additional classification information. We set up the following experiments to evaluate this hypothesis, using the results of MAIRA-1 as the baseline: 1)Using LLM combined with classification information to refine the reports generated by one (referred to as *Re. Single*) or multiple (referred to as *Re. Multi*) pre-trained report generation models. In this setup, we use No. 3 in Table 8 as the single model and both No. 3 and No. 6 in Table 8 as the multiple models to generate the reports to be refined; 2) Using only classification information, LLM iteratively generate and refine the generated reports, referred to as *Iteration n*, where $n$ represents the number of iterations. We use Llama 3.1 70B as the default LLM, and the classification prompts are generated from the outputs of the Swin-Transformer-L trained on the MIMIC-CXR dataset. The results, shown in Table A, indicate that using classification information to refine based on the reports generated by the pre-trained models leads to improvements in most metrics. Notably, when comparing *Re. Multi* with the *Baseline*, the classification metric Macro-F1-14 improves by 2.7%. Meanwhile, the metrics of generated reports using only classification information tend to decline with the increase in refining iterations.

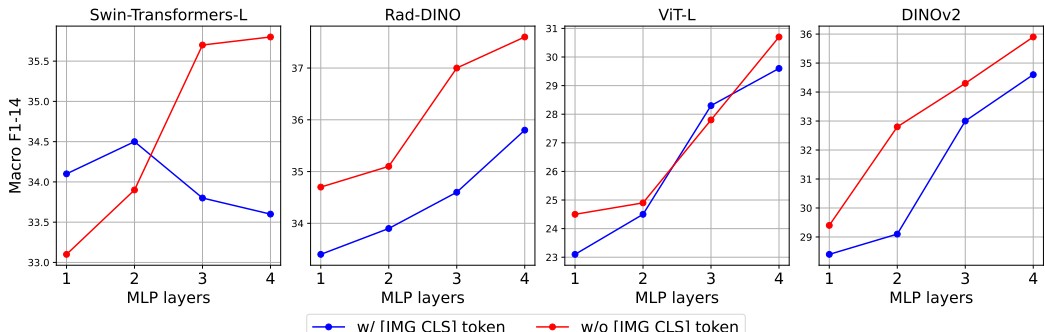

Figure 2: **The impact of the *[CLS]* token on the classification performance of report generation.** We evaluated whether the *[CLS]* token carried by four pre-trained vision encoders on the MIMIC-CXR dataset affects the 14-class classification performance of the generated report. The results show that the *[CLS]* token from the image does not significantly improve the classification performance of the generated report.

**Variation of vision encoder.** We evaluate the impact of four vision encoders on report generation using the MIMIC-CXR dataset. Specifically, we assess the performance of Swin Transformers (Liu et al., 2021), Rad-DINO (Pérez-García et al., 2024), ViT-L (Dosovitskiy et al., 2021), and

Table 3: The impact of adding the different format of classification information.

(a) The baseline refers to using only Rad-DINO as the classifier. T and I represent text and image *[CLS]* tokens, respectively. The symbol † indicates the classification performance measured on the generated report, while the absence of this symbol indicates the average classification performance measured only on the classification head. ∗ represents multiple classification tokens.

| Metrics | Baseline | T+I [CLS] | T+I [CLS]$^{\dagger}$ | T$^*$+I [CLS] | T$^*$+I [CLS]$^{\dagger}$ |
|---|---|---|---|---|---|
| Macro-F1-14 | 57.1 | 62.5 / 60.7 | 35.8 | 63.8 / 60.7 | 33.5 |
| Micro-F1-14 | 60.9 | 65.6 / 63.1 | 49.5 | 66.2 / 63.1 | 47.1 |

(b) Use the output of Rad-DINO as the classification prompt. The Baseline refers to not using any classification information. **ALL** means converting all classification information into prompts. **Prob.** indicates attaching the corresponding classification probability. **Only Pos.** means only converting observations that are positive.

| Metrics | Baseline | ALL | ALL + Prob. | Only Pos. | Only Pos. + Prob. |
|---|---|---|---|---|---|
| Macro-F1-14 | 36.7 | 38.8 | 38.6 | 37.9 | 38.0 |
| Micro-F1-14 | 54.6 | 56.5 | 56.1 | 55.3 | 55.9 |

Table 4: The zero-shot results of long-tail classifiers on different methods on the IU X-Ray dataset. 'C' represents the classifier, 'V' is the vision encoder. The **bold** indicates the best value. 🔥 and ❄️ indicates whether the backbone is trainable or frozen, respectively.

| Method | ROUGE-L | BLEU-1/-4 | METEOR | RJ-1 | RJ-2 | RJ-3 | RJ-4 | Macro-F1-OOD80 |
|---|---|---|---|---|---|---|---|---|
| LT-200 Classifier | - | - | - | - | - | - | - | 13.6 |
| 🔥V + LLM | **22.5** | 33.5 / **7.6** | 29.0 | **0.15** | **2.30** | **0.07** | 0.93 | 8.3 |
| 🔥C + V + LLM | **22.5** | **33.6 / 7.6** | **29.1** | **0.15** | 2.33 | 0.07 | 0.85 | **9.3** |
| ❄️Expanding | 15.7 | 23.2 / 3.9 | 22.5 | 0.25 | 2.42 | 0.09 | **0.78** | 3.5 |
| ❄️Refining | 19.3 | 26.7 / 4.9 | 28.2 | 0.16 | 2.23 | **0.05** | 0.91 | 6.5 |

DINOv2 (Oquab et al., 2024). As shown in Fig 2, the results indicate that the choice of vision encoder affects both classification and report generation performance.

**The scales of LLM.** We conduct experiments to assess the impact of the LLM's scales to various baselines. By default, we set the classifier to Swin Transformers-Large, and the vision encoder to Rad-DINO. In the fine-tuning paradigm, we use Phi-3 3B, Vicuna-1.5 7B/13B, and Llama3.1 7B/13B as the large language models (LLMs). In the prompt learning paradigm, without introducing additional vision information, we employ prompt engineering to evaluate the effect of classification information on report quality. We compare two types of prompts: in-context learning (ICL) prompts and directly inputting classification information into the LLM to generate the corresponding reports. The format of the ICL prompt is as follows: *Here is the classifier result for this Chest X-ray: [...], and the corresponding report is: [...]. Now, based on the classification result of a new Chest X-ray image: [...], provide a reasonable and rigorous report.*

The results, presented in Table 8, indicate that comparisons across different scales of baselines show that using classification as additional information can improve in-domain performance. For instance, in the Macro-F1 classification, No. 3 versus No. 4 showed a significant improvement of 2%. However, the results for ROUGE-L and BLEU-4 metrics were worse when compared to smaller models, such as RGRG. Additionally, we find that Experiment 1 yield similar performance to the baseline. Experiments 3-5 suggest that increasing the scale of the LLM can enhance language metrics performance but offers limited improvement in classification performance. Experiments 9-12 demonstrate that the absence of vision information leads to a significant decline in overall performance.

## 3.3 THE DIFFERENCES BETWEEN OOD AND IN-DOMAIN SCENARIOS

**Can the long-tail classifier help the generation model on out-of-domain's long-tail data?** We conducted comparative experiments on two OOD datasets (PadChest and IU X-ray). We first se-

Table 5: The zero-shot results of long-tail classifiers on different methods on the PadChest dataset. 'C' represents the classifier, 'V' is the vision encoder. The **bold** indicates the best value. 🔥 and ❄️ indicates whether the backbone is trainable or frozen, respectively.

| Method | ROUGE-L | BLEU-1/-4 | METEOR | RJ-1 | RJ-2 | RJ-3 | RJ-4 | Macro-F1-OOD99 |
|---|---|---|---|---|---|---|---|---|
| LT-200 Classifier | - | - | - | - | - | - | - | 12.8 |
| 🔥V + LLM | 14.9 | 17.8 / **1.8** | 19.6 | 0.41 | **1.54** | 0.09 | 0.10 | **7.5** |
| 🔥C + V + LLM | **15.5** | **18.3** / **1.8** | **20.4** | 0.46 | 1.55 | 0.09 | 0.08 | 7.4 |
| ❄️Expanding | 13.9 | 16.8 / 1.2 | 19.7 | 0.39 | 2.19 | **0.08** | **0.07** | 3.5 |
| ❄️Refining | 15.7 | 17.1 / 1.6 | 19.9 | **0.38** | 2.07 | 0.08 | **0.08** | 5.3 |

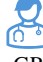
ID: 170

LT Classifier: Pleural Effusion, Cardiomegaly, Edema, and Lung Opacity are positive. Mediastinal Contour and Bony are negative.
GT: Opacity, Pulmonary, Atelectasis, and Pleural Effusion are positive.

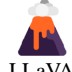
GR

Patchy left basilar subsegmental atelectasis, infiltrates and/or small left pleural effusion. The cardiac silhouette is at the upper limits of normal for size. Patchy opacities are demonstrated in the left lung base. No focal pulmonary consolidation. No pneumothorax. Minimal degenerative changes of the thoracic spine.

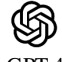
LLaVA

PA and lateral views of the chest were provided. There is left lower lobe consolidation, compatible with pneumonia. There is a small left pleural effusion. There is mild pulmonary edema. The heart size is difficult to assess. Mediastinal contour is normal. Bony structures are intact.

GPT-4

PA and lateral views of the chest were provided. There is left lower lobe consolidation, compatible with pneumonia, although the classification indicates no pneumonia. There is a small left pleural effusion, which is consistent with the classification. There is mild pulmonary edema, which aligns with the classification results. The heart size is difficult to assess, but cardiomegaly is indicated in the classification. Mediastinal contour is normal. Bony structures are intact.

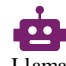
Llama

PA and lateral chest radiographs are available. Left lower lobe consolidation consistent with pneumonia. Small effusion in the left pleural cavity. Mild pulmonary edema. Cardiac enlargement. Normal mediastinal contour. Bone structure intact.

Figure 3: We present a sample report from the IU X-Ray test set, and the reports generated by the three baseline models. GR and GT represent the ground-truth report and ground-truth observations, respectively. The classification results come from LT-200 classifier. The green text indicates that this observation appeared in all reports, while the purple text indicates that this observation was not mentioned in the ground truth but appeared in the report.

lected four baselines to generate corresponding reports based on the LT-200 classification results for the long-tail classifier. These baselines were categorized by trainable ('V+LLM' and 'C+V+LLM') and frozen weights('Expanding' and 'Refining'). We reported results for language metrics and LLM-based metrics, as shown in Table 4 and Table 5. The results indicate that the trainable models like 'C+V+LLM' consistently outperform frozen models, but all models struggle with long-tail classification, highlighting the challenges LLM-based models face with OOD data.

To further investigate this phenomenon, we conducted extensive case studies[3], the partial result as shown in Fig. 3. We find that significant discrepancies between the actual diagnostic report and the report generated by the the model that integrated the long-tail classification information. The GT diagnostic report primarily emphasizes atelectasis, pleural effusion, and pulmonary opacity, whereas the generative reports erroneously identifies cardiomegaly and edema as positive findings, neglecting atelectasis altogether. Additionally, while the pleural effusion noted in the LT Classifier report aligns with the true diagnosis, the false positives regarding cardiomegaly and edema may be attributed to biases in the training data or potential overfitting of the model.

**Finding 2:** *The long-tail classifier offers limited assistance for report generation in out-of-distribution (OOD), constrained by the generalization performance of classifier on OOD issues.*

---

[3]More cases are shown in the Appendix.

## 4 RELATED WORK

### 4.1 AUTOMATIC REPORT GENERATION

Automatic report generation has gained attention in healthcare and NLP, with various approaches leveraging NLG and deep learning (Jing et al., 2017; Bannur et al., 2024; Jin et al., 2024; Tu et al., 2024). Early methods used retrieval-based (Li et al., 2019) and template-based techniques (Biswal et al., 2020; Harzig et al., 2019; Li et al., 2018), which lacked flexibility. Advances in large language models (LLMs) (Bannur et al., 2024; Hyland et al., 2023; Wang et al., 2023; Zhao et al., 2024b) have enabled more sophisticated systems, showing improved coherence in report generation (Li et al., 2024; Zhao et al., 2024a). Multi-modal data integration, combining text and images, further enhances report interpretability. However, diagnostic accuracy remains an issue compared to traditional methods, prompting efforts (Jin et al., 2024; Zhao et al., 2024b; Wang et al., 2023) to improve accuracy by incorporating diagnostic imaging. More discussions are present at the appendix.

### 4.2 RADIOLOGY REPORT EVALUATION

Radiology report evaluation focuses on accuracy and clinical relevance, assessed via language and clinical metrics. Language metrics, like BLEU (Papineni et al., 2002), ROUGE (Lin, 2004), METEOR (Banerjee & Lavie, 2005), and BERTScore (Zhang et al., 2019), measure similarity but lack clinical depth. Clinical metrics assess medical events, with tools like CheXpert and CheXbert (Smit et al., 2020), RadGraph (Jain et al., 2021), and cosine similarity, though limited by predefined entities. Recent methods, including RadCliQ (Yu et al., 2023), RadEval (Calamida et al., 2023), and LLM-based approaches (Wang et al., 2024), show improved adaptability and performance. More discussions are present at the appendix.

## 5 DISCUSSION AND LIMITATIONS

**The gap between report generation and classification.** Our experiments reveal a significant gap between report generation and classification, both in in-domain and OOD long-tail scenarios. Classification only requires determining whether an observation is positive, while report generation demands detailed text that mirrors the target report, including specifics like location and severity. The absence of descriptive details in classification can cause hallucinations in LLM methods, leading to poor reports. Some studies (Wang et al., 2023; Zhao et al., 2024b) have improved report quality by using more complex information. A promising approach is to enable LLMs to selectively use additional data during fine-tuning, such as dynamically adjusting attention weights (Chefer et al., 2023; Zhou et al., 2024; Liu et al., 2024b).

**Evaluation Framework.** Developing a robust evaluation framework for free-text reports remains a challenge. Current language and clinical metrics are inadequate: language metrics focus on grammatical similarities but miss the precision required for clinical diagnostics, while clinical metrics are too narrow to capture the diverse scenarios in medical reports. In this paper, we enhance both using LLMs, as they can interpret complex texts and support knowledge extrapolation. However, this has limitations, especially in OOD scenarios, where varying granularity in observation names across datasets requires semantic transformation. In long-tail datasets, overlapping observations can further reduce the reliability of LLM-based metrics.

## 6 CONCLUSION

In this study, we explore and understand how diagnossis results impact the overall quality of LLM-based report generation models. We design a long-tail evaluation framework that incorporates both in-domain and out-of-domain (OOD) elements, utilizing LLM-based language metrics and clinical metrics. Furthermore, we conducted a high-level analysis of the effective combinations of the primary components of LLM-based generation models, assessing how classification information impacts report quality across four benchmarks. Our findings reveal that the classifier's performance in long-tail observations directly influences the overall performance of the LLM-based generation model. We hope these findings inspire further enthusiasm for more robust report evaluation metrics and more effective report generation models.

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

# APPENDICES

## A THE GAP BETWEEN GENERATION AND CLASSIFICATION

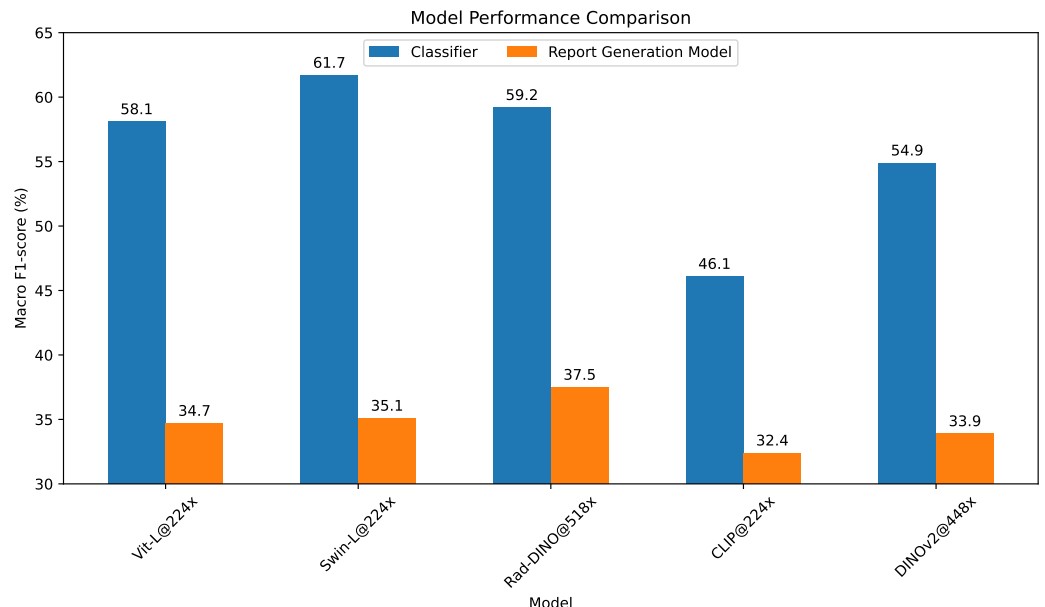

Figure 4: **The gap between generation and classification.**

We utilized five mainstream backbones on the MIMIC-CXR dataset to evaluate their performance on both classification and report generation tasks. For the classification task, ViT-L and Swin-L were fine-tuned on MIMIC-CXR, while the other three backbones had their weights frozen and were followed by a linear classification head. For the report generation task, we replaced the classification head of each backbone with the vision encoder module from LLaVA. As shown in Fig. 4, the results indicate that classification outperforms report generation by approximately 20% in terms of Macro F1-score.

## B MODEL DESIGN SPACE

See Table 6.

## C THE PERFORMANCE OF REFINING REPORT

See Table 7.

Table 6: Model design space.

| No. | Vision Encoder | Classifier | LLM |
|-----|----------------|------------|-----|
| 1 | - | Swin Transformer-Large | Phi-3 3B |
| 2 | - | Swin Transformer-Large | Vicuna1.5 7B |
| 3 | - | Swin Transformer-Large | Vicuna1.5 13B |
| 4 | - | Swin Transformer-Large | Llama3.1 7B |
| 5 | - | Swin Transformer-Large | Llama3.1 13B |
| 6 | - | Rad-DINO | Phi-3 3B |
| 7 | - | Rad-DINO | Vicuna1.5 7B |
| 8 | - | Rad-DINO | Vicuna1.5 13B |
| 9 | - | Rad-DINO | Llama3.1 7B |
| 10 | - | Rad-DINO | Llama3.1 13B |
| 11 | Swin Transformer-Large | - | Phi-3 3B |
| 12 | Swin Transformer-Large | - | Vicuna1.5 7B |
| 13 | Swin Transformer-Large | - | Vicuna1.5 13B |
| 14 | Swin Transformer-Large | - | Llama3.1 7B |
| 15 | Swin Transformer-Large | - | Llama3.1 13B |
| 16 | Rad-DINO | - | Phi-3 3B |
| 17 | Rad-DINO | - | Vicuna1.5 7B |
| 18 | Rad-DINO | - | Vicuna1.5 13B |
| 19 | Rad-DINO | - | Llama3.1 7B |
| 20 | Rad-DINO | - | Llama3.1 13B |
| 21 | Swin Transformer-Large | Swin Transformer-Large | Phi-3 3B |
| 22 | Swin Transformer-Large | Swin Transformer-Large | Vicuna1.5 7B |
| 23 | Swin Transformer-Large | Swin Transformer-Large | Vicuna1.5 13B |
| 24 | Swin Transformer-Large | Swin Transformer-Large | Llama3.1 7B |
| 25 | Swin Transformer-Large | Swin Transformer-Large | Llama3.1 13B |
| 26 | Rad-DINO | Swin Transformer-Large | Phi-3 3B |
| 27 | Rad-DINO | Swin Transformer-Large | Vicuna1.5 7B |
| 28 | Rad-DINO | Swin Transformer-Large | Vicuna1.5 13B |
| 29 | Rad-DINO | Swin Transformer-Large | Llama3.1 7B |
| 30 | Rad-DINO | Swin Transformer-Large | Llama3.1 13B |

Table 7: Comparison of results for different methods of refining generated reports using classification information

| Exp. | ROUGE-L | BLUE-1/-4 | METEOR | $RG_{ER}$ | Macro-F1-14 | Macro-F1-5 |
|------|---------|-----------|--------|-----------|-------------|------------|
| Baseline | 29.8 | 37.7 / 14.6 | 33.2 | 29.0 | 36.7 | 46.1 |
| Re. Single | 30.1 | 38.5 / 16.0 | 33.4 | 29.7 | 38.2 | 46.4 |
| Re. Multi. | 27.4 | 41.0 / 14.7 | 33.7 | 29.1 | 39.4 | 49.0 |
| Iteration 1 | 19.8 | 27.5 / 5.5 | 22.7 | 19.2 | 25.5 | 38.6 |
| Iteration 3 | 13.5 | 16.1 / 3.8 | 18.0 | 12.5 | 14.9 | 21.0 |

## D  THE SCALES OF LLMS

See Table 8.

## E  RELATED WORK

### E.1  AUTOMATIC REPORT GENERATION

Automatic report generation has garnered significant attention in recent years, particularly in fields such as healthcare and natural language processing. Researchers (Jing et al., 2017; Bannur et al., 2024; Jin et al., 2024; Tu et al., 2024) have investigated various methods to automate report creation, leveraging techniques from natural language generation (NLG) and deep learning. Early approaches

Table 8: We report the performance differences of various models on the MIMIC-CXR dataset with and without the use of the classifier. † indicates that the result is directed cited from the original paper. § denotes that adding classification prompts into the model's input.

| NO. | Method | Params. (B) | ROUGE-L | BLUE-1/-4 | METEOR | $RG_{ER}$ | Macro-F1-14 | Micro-F1-14 | Macro-F1-5 | Micro-F1-5 |
|---|---|---|---|---|---|---|---|---|---|---|
| | 🔥*Fine-tuned (V: Rad-DINO)* | | | | | | | | | |
| 1 | Phi-3-3B | 4 | 29.9 | 35.3 / 14.1 | 32.3 | 27.9 | 34.7 | 53.6 | 46.2 | 54.9 |
| 2 | Phi-3-3B§ | 4 | 29.8 | 36.4 / 13.9 | 32.5 | 27.9 | 36.5 | 55.1 | 47.6 | 55.1 |
| 3 | Vicuna-1.5-7B | 7 | 29.8 | 37.7 / 14.6 | 33.2 | 29.0 | 36.7 | 54.6 | 46.1 | 54.8 |
| 4 | Vicuna-1.5-7B§ | 7 | 29.9 | 34.8 / 13.6 | 31.8 | 28.0 | 38.7 | 53.1 | 48.9 | 55.5 |
| 5 | Vicuna-1.5-13B§ | 13 | 30.0 | 36.1 / 14.0 | 32.2 | 27.9 | 38.9 | 54.4 | 49.1 | 55.6 |
| 6 | Llama3.1-8B | 8 | 29.6 | 37.8 / 14.7 | 33.1 | 28.9 | 36.6 | 53.9 | 46.0 | 54.6 |
| 7 | Llama3.1-8B§ | 8 | 29.9 | 35.6 / 14.1 | 32.6 | 27.9 | 38.1 | 55.7 | 48.8 | 56.1 |
| 8 | Llama3.1-13B§ | 13 | 30.0 | 36.1 / 14.0 | 32.2 | 27.9 | 38.9 | 54.4 | 49.1 | 55.6 |
| | ❄️*Prompt Learning* | | | | | | | | | |
| 9 | ICL + Llam3.1-8B§ | 8 | 19.5 | 26.7 / 4.5 | 22.7 | 19.1 | 21.4 | 34.2 | 34.6 | 43.7 |
| 10 | Llam3.1-8B§ | 8 | 19.4 | 26.6 / 4.3 | 22.3 | 18.9 | 25.0 | 42.2 | 38.1 | 48.2 |
| 11 | ICL + Llam3.1-70B§ | 70 | 19.6 | 27.1 / 5.0 | 22.6 | 19.1 | 22.8 | 34.3 | 35.2 | 43.9 |
| 12 | Llam3.1-70B§ | 70 | 19.8 | 27.5 / 5.5 | 22.7 | 19.2 | 25.5 | 43.9 | 38.6 | 48.9 |

primarily employed retrieval-based (Li et al., 2019) and template-based methods (Biswal et al., 2020; Harzig et al., 2019; Li et al., 2018) , where predefined structures were populated with relevant data. However, these methods often lacked flexibility and could not adapt to varying contexts. Recent advancements in deep learning, especially the use of large language models (LLM) (Bannur et al., 2024; Hyland et al., 2023; Wang et al., 2023; Zhao et al., 2024b), have facilitated the development of more sophisticated report generation systems. Studies (Li et al., 2024; Zhao et al., 2024a) have demonstrated the effectiveness of powerful foundation models in generating coherent and contextually relevant reports from structured data inputs. Furthermore, the integration of multi-modal data, such as images and text, has shown promise in enhancing the richness and interpretability of generated reports.

Despite these advancements, the diagnostic accuracy of reports generated by these advanced models still exhibits significant performance gaps compared to traditional medical image classification. Consequently, numerous efforts (Jin et al., 2024; Zhao et al., 2024b; Wang et al., 2023) have aimed to incorporate diagnostic results from imaging to enhance the accuracy of generated reports. This article systematically summarizes the potential combinations of these approaches and conducts a comprehensive evaluation of their effectiveness in addressing both in-domain and out-of-domain long-tail issues.

### E.2 RADIOLOGY REPORT EVALUATION

The evaluation of radiology reports is essential for ensuring their accuracy, completeness, and clinical relevance. Traditional evaluation methods can be categorized into two main types: language metrics and clinical metrics. Language metrics include BLEU (Papineni et al., 2002), ROUGE (Lin, 2004), and METEOR (Banerjee & Lavie, 2005) scores, as well as more recent metrics like BERTScore (Zhang et al., 2019), which utilize embeddings from pre-trained models. These metrics are commonly employed to assess the similarity between generated reports and ground-truth reports. However, they often overlook the clinical events described in radiology reports, resulting in limited clinical significance.

On the other hand, clinical metrics focus on the clinical descriptions within radiology reports, which are vital for practical applications. These metrics capture all clinical events illustrated in medical images, such as the nature, location, and extent of pathology. A widely used metric is CheXpert, which categorizes 14 types of pathologies and indicates their presence or absence with labels. Tools like CheXbert (Smit et al., 2020), along with metrics such as cosine similarity and RadGraph (Jain et al., 2021), are employed for this evaluation. However, these extraction-based methods have limitations due to their dependence on predefined entities and strict matching rules. Efforts to enhance these methods, such as RadCliQ (Yu et al., 2023) and RadEval (Calamida et al., 2023), which combine different metrics—still struggle to fully evaluate clinical descriptions. Recently, an innovative approach (Wang et al., 2024) that leverages large language models for assessment offers improved adaptability and performance comparable to that of radiologists.

## F   IMPLEMENTATION DETAILS

Following (Liu et al., 2024a), we adopt hyper-parameters similar to those in LLaVA-1.5 for training, jointly tuning the LLM with a randomly initialized the vision connector. We do not include a precursor training step to pretrain the adapter, as pretraining provides no significant performance improvement (Hyland et al., 2023). The model is trained for 3 epochs without using parameter-efficient fine-tuning techniques. We use AdamW as the optimizer, with $\beta_1$ and $\beta_2$ set to 0.9 and 0.999, respectively. The global batch size for all models is set to 128. A cosine learning rate scheduler with a warm-up factor of 0.03 and an initial learning rate of $2 \times 10^{-5}$ is used for learning rate adjustment. All experiments are conduct on a cluster of 4 NVIDIA A100 80G GPUs.

## G   MORE CASE STUDIES

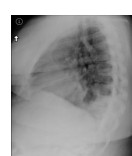

ID: 138

LT Classifier: Atelectasis and Pleural Effusion are positive. Mediastinal Widening and Pneumothorax are negative.
GT: Pleural Effusion, Pulmonary Atelectasis, and Lung Opacity are positive.

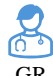

GR

Pulmonary detail obscured secondary to body habitus and superimposed soft tissue. **Again seen is a left basilar opacity compatible with some effusion and adjacent left basilar atelectasis.** Overall, size of effusion appears slightly smaller. Right lung stable and grossly clear. No XXXX acute abnormalities since the previous chest radiograph.

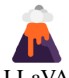

LLaVA

A right IJ central venous catheter is seen with its tip residing at the level of the low SVC. There is left basilar opacity compatible with effusion and likely also atelectasis. The right lung appears clear. No pneumothorax is seen. The heart size appears grossly within normal limits. The mediastinal contour is unremarkable. The bony structures are intact.

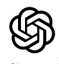

GPT-4

A right IJ central venous catheter is seen with its tip residing at the level of the low SVC, confirmed as negative for any associated complications. There is left basilar opacity compatible with effusion and likely also atelectasis, both of which are positive findings. The right lung appears clear, with no evidence of any abnormalities. No pneumothorax is present. The heart size appears grossly within normal limits, with no cardiomegaly. The mediastinal contour is unremarkable, showing no abnormalities. The bony structures are intact, with no fractures or other bony abnormalities detected.

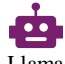

Llama

A right IJ central venous catheter is visible with its tip at the level of the inferior SVC. The left lung base is opaque, consistent with an effusion and possible atelectasis. The right lung appears clear. No pneumothorax is seen. The heart size appears within normal limits. The mediastinal contour is unremarkable. The bone architecture is intact.

Figure 5: We present a sample report from the IU X-Ray test set, and the reports generated by the three baseline models. GR and GT represent the ground-truth report and ground-truth observations, respectively. The classification results come from LT-200 classifier. The green text indicates that this observation appeared in all reports, while the purple text indicates that this observation was not mentioned in the ground truth but appeared in the report.

