# OpenReview forum: "Is classification all you need for radiology report generation?"
_ICLR.cc/2025/Conference — Submitted to ICLR 2025_

### Official Review · Reviewer_91mZ · 2024-11-01

**Soundness:** 2
**Presentation:** 3
**Contribution:** 2
**Rating:** 6
**Confidence:** 5

**Summary:**

The paper explores the use of classification models in enhancing radiology report generation, specifically examining if a classifier encompassing all possible radiological findings can improve the robustness of generated reports. The study finds that while classification models improve report quality in in-domain scenarios, their impact is limited in out-of-distribution (OOD) settings, especially for long-tail disease categories. The authors propose a comprehensive evaluation framework with novel LLM-based metrics and conduct experiments across multiple benchmarks, concluding that classification aids in in-domain accuracy but may introduce errors in OOD cases due to model biases.

**Strengths:**

The study tackles a critical challenge in medical AI—how to generate reliable and clinically relevant radiology reports that generalize well across diverse clinical scenarios. By showing that classification models can enhance in-domain performance while introducing potential errors in OOD settings, this paper offers valuable insights into the limitations and trade-offs of using classification-aided generation models. The paper is well-structured, with each section building logically upon the previous one. The problem formulation, methodology, and experimental setup are clearly articulated, enhancing accessibility for readers. The experimental design is thorough, incorporating multiple benchmarks and a diverse set of evaluation metrics, including novel LLM-based metrics, to assess model performance across both in-domain and OOD settings. The rigorous use of benchmarks and detailed comparisons with existing methods underscore the study's robustness and the reliability of its findings.

**Weaknesses:**

The primary weakness of this work lies in its motivation, or key argument, which has already been discussed in a prior publication that is not referenced in this submission. The previous study, Medical Report Generation Is A Multi-label Classification Problem (Fan et al., IEEE MedAI 2024), introduces a similar concept: that classification accuracy significantly impacts the quality of medical report generation. Both works share the same foundational idea, though they differ in specific model designs and classification categories. From a performance perspective, the previous work, which uses a cross-attention approach, demonstrates better outcomes and introduces the concept of ceiling performance. This concept suggests that the theoretical upper bound for this approach would be achieved if all ground-truth labels were provided to the model, given that real-world classifiers cannot reach 100% accuracy.

While this submission’s strength lies in its extensive experiments, broader metrics, and a variety of models, its core insights are similar to the prior work, without addressing ceiling performance or providing a comparison to it. Additionally, the reason why the 28 classification categories used in the previous study outperform disease-based categories is not examined here. These points, along with further experimental and discussion-based additions, would significantly enhance the paper.

**Questions:**

please read the weakness part.

---

> ### Author Response · Authors · 2024-11-27
> **Response to Reviewer 91mZ.**
>
> R: We respectfully disagree with this comment.
>
> **Comparison from the perspective of motivation:**
> The compared work "Medical Report Generation Is A Multi-label Classification Problem" investigates a specific classification-based approach, reaching conclusions similar to ours. However, this scenario represents only a subset of our study’s scope, as we also consider OOD scenarios. Additionally, we have observed a significant gap between diagnosis performance and report generation performance.
>
> **Comparison from the perspective of methodology:**
> Our approach emphasizes high-level design paradigms and investigates how diagnosis influences report generation. We explore four different design paradigms from existing methods, while the compared approach essentially falls under one of our paradigms. Moreover, the compared work focuses more on algorithmic design details, whereas we focus on the broader impact of diagnosis on report generation.
>
> **Comparison from the perspective of experiments:**
> We conducted extensive experiments across various classifiers, vision encoders, and LLMs. Our findings reveal that classification impacts report generation in different ways across in-domain, long-tailed, and OOD scenarios. In contrast, the compared work does not consider these experimental settings.
>
> Finally, we are currently unable to conduct a direct comparison with this work because it does not provide reproducible source code. Furthermore, we were unable to locate this article in the IEEE MedAI repository.
>
> We sincerely request the reviewer to re-evaluate our submission to ensure a fair and accurate assessment.

---

> > ### Comment · Reviewer_91mZ · 2024-11-29
> > **response to authors**
> >
> > I appreciate your clarification and agree with some of the points made, particularly that the compared approach represents a subset of your study’s scope. However, I maintain my concerns regarding the novelty and conceptual contributions of your work, as outlined below:
> >
> > Both your submission and the compared work emphasize the significant influence of classification accuracy on report generation quality. While your study extends the experimental scope, the core idea—classification impacts report generation—remains largely similar. I have not observed substantially novel concepts or methodologies beyond an expanded set of experiments.
> >
> > The compared work leverages the knowledge graph nodes from Wang et al [1]. as classification targets, which has become a widely used auxiliary tool in MRG research. This integration has demonstrated effectiveness in improving classification and report generation. However, your work does not include this important baseline or a similar implementation. The KG’s 28 categories are not fundamentally different from your long-tailed 200 categories in terms of classification objectives, yet the omission of this established approach weakens the experimental comparison.
> >
> > [1] Zhang, Yixiao, et al. "When radiology report generation meets knowledge graph." Proceedings of the AAAI conference on artificial intelligence. Vol. 34. No. 07. 2020.
> >
> > While your study claims to address OOD scenarios, it’s important to note that existing MRG datasets are predominantly single-center, and OOD settings remain an uncommon use case in practice. Additionally, your findings indicate that the impact of OOD scenarios on report generation is minimal, which reduces the significance of this claimed contribution.
> >
> > You mentioned that the compared approach does not provide reproducible code. However, implementing classification experiments using the KG nodes as categories (as in the compared work) does not appear overly challenging, particularly given that this methodology has been widely adopted in subsequent research. Furthermore, your work uses LLMs combined with classification results as a baseline but fails to include R2GenGPT [2], which represents the most comparable baseline in this context. This omission is notable and weakens your experimental comparisons.
> >
> > [2] Wang, Zhanyu, et al. "R2gengpt: Radiology report generation with frozen llms." Meta-Radiology 1.3 (2023): 100033.
> >
> > In conclusion, while I recognize the value of your extended experiments and the broader scope of your analysis, I believe the conceptual novelty of your work remains limited. Without more substantial differentiation from existing methods, the manuscript does not sufficiently advance the state of knowledge in MRG. I encourage the authors to address these concerns and incorporate the discussed baselines for a more robust comparison. More discussions are welcome.

---

> ### Author Response · Authors · 2024-12-02
> **Response to reviewer 91mZ.**
>
> Thank you for your feedback. The method [1] you provided represents an important baseline. However, we believe that methods based on knowledge graphs (KGs) are fundamentally similar to one of our baselines, "Expanding." Both approaches rely on classification and predefined templates to generate the final report. In future work, we will explore further possibilities for improving the "Expanding" baseline, such as fine-tuning. Additionally, our method has been evaluated on a larger dataset compared to [1].
> Moreover, our primary focus is on baselines centered on large language model (LLM)-based report generation. The conclusions of [1], while noteworthy, are significantly limited in the context of current technological advancements. Nevertheless, we highly respect these contributions and will consider refining our methodology to integrate these insights.
>
> We respectfully disagree with your comment that Out-of-domain (OOD) case represents an uncommon case. OOD data is far from rare; instead, it is a critical challenge, particularly in deep learning-driven medical imaging analysis. This is because the distribution of medical imaging data can be influenced by various factors, including differences in patient populations, variations in imaging devices and protocols, the emergence of new diseases, and fluctuations in image quality[3-5].
>
> As for the literature you provided [2], it closely aligns with our baseline (V+L) and demonstrates similar or slightly mismatched performance on the same dataset, MIMIC-CXR. For example, the metrics indicate comparable or marginally differing results: BLEU-4 (V+L: 14.6 vs. [2]: 13.4), METEOR (V+L: 32.3 vs. [2]: 16.0), and F1-14 (V+L: 35.7 vs. [2]: 35.8).
>
> [1] Zhang, Yixiao, et al. "When radiology report generation meets knowledge graph." Proceedings of the AAAI conference on artificial intelligence. Vol. 34. No. 07. 2020.
>
> [2] Wang, Zhanyu, et al. "R2gengpt: Radiology report generation with frozen llms." Meta-Radiology 1.3 (2023): 100033.
>
> [3] Mehrtash, Alireza, et al. "Confidence calibration and predictive uncertainty estimation for deep medical image segmentation." IEEE transactions on medical imaging 39.12 (2020): 3868-3878.
>
> [4] Narayanaswamy, Vivek, et al. "Exploring inlier and outlier specification for improved medical ood detection." Proceedings of the IEEE/CVF International Conference on Computer Vision. 2023.
>
> [5] Yuan, Mingze, et al. "Devil is in the queries: advancing mask transformers for real-world medical image segmentation and out-of-distribution localization." Proceedings of the IEEE/CVF Conference on Computer Vision and Pattern Recognition. 2023.

---

> > ### Comment · Reviewer_91mZ · 2024-12-02
> > **response to authors**
> >
> > Thank you for your response. I have revisited your work and compared it with [1], and I acknowledge the added value of your study in emphasizing classification performance and analyzing OOD scenarios. I appreciate your explanation regarding the importance of OOD challenges in medical imaging analysis. However, my concern stems from the fact that existing MRG benchmarks, such as MIMIC-CXR, are inherently drawn from single-center data and thus may not fully reflect the challenges of OOD distributions. That said, I agree that your work could provide valuable guidance for future applications of LLMs in broader and more diverse MRG tasks. Highlighting this potential impact could further enhance the relevance of your findings.
> >
> > While [1] does discuss the significance of classification performance, your study offers richer experiments and more diverse settings. The experimental results in your work are more robust and provide solid evidence to support your conclusions. However, I suggest explicitly discussing [1] in the revision, particularly in the context of how your findings build upon and extend its contributions. Additionally, expanding the “Expanding” baseline with experiments that incorporate insights from [1], such as KG-based classification enhancements, could further strengthen this aspect of your study. Given the relevance of R2GenGPT [2] as a baseline for LLM-based MRG, I recommend including its results in your comparisons. Furthermore, analyzing the reasons behind specific metric differences, such as the notable discrepancy in METEOR versus BLEU-4 scores, could provide valuable insights into the unique strengths of your approach. This analysis would not only clarify performance differences but also help demonstrate the broader applicability of your findings.
> >
> > I believe that incorporating these two key revisions—(1) a discussion of [1] and an expanded “Expanding” baseline, and (2) a comparison with R2GenGPT and an analysis of metric differences—will enhance the completeness and impact of your work. I hope these suggestions will be addressed in the final revision, I would be happy to revise my score to a 6, as your work can advance MRG research.

---

> > > ### Author Response · Authors · 2024-12-03
> > > **Thank you for your response and recognition.**
> > >
> > > Thank you very much for your response and recognition of our work. These positive comments will undoubtedly serve as one of the driving forces for our future works. In the revision, we will carefully discuss the relationship between the conclusions of references [1, 2] and our method. Once again, we sincerely thank you for acknowledging the contributions our work has made in the MRG field.

---

### Official Review · Reviewer_ueGE · 2024-11-01

**Soundness:** 3
**Presentation:** 3
**Contribution:** 2
**Rating:** 5
**Confidence:** 4

**Summary:**

Current RRG systems exhibit a notable gap in clinical metrics when compared to classification models. The authors investigate whether training a classifier that encompasses all possible long-tailed and rare diseases could enhance the robustness of reports. Key findings show that classification helps improve report quality in in-distribution settings but exhibits limited benefits in OOD scenarios.

**Strengths:**

1. The paper evaluates whether classification models can help improve the quality of radiology reports. This is a novel and interesting insight, with potential for aiding in real-world clinical workflows.
2. The authors perform extensive evaluations across different strategies for incorporating classification information as well as in-domain vs. OOD settings.
3. The authors also introduce novel LLM-based metrics for assessing the quality of radiology reports, which evaluate how well reports capture long-tail observations. This metric is an interesting contribution and has potential for aiding future works in the domain of radiology report generation.

**Weaknesses:**

The key weakness of this paper is inadequate evaluations, as discussed below.

1. **Inadequate evaluations of classifier:** The incorporation of the classifier is inadequately evaluated, making it difficult to understand the settings in which incorporating a classifier is useful, as detailed below. The paper could have benefitted from more nuanced insights on when including the classifier helps vs. detracts from the quality of generated reports.

    a. **Classifier Performance:** Although the incorporation of the classifier is the key contribution of this paper, the authors do not provide any evaluations on the quality of the actual classifier. What conditions does the classifier perform well on and which conditions does the classifier perform worse on? Does this correlate with report generation performance on the subset of images with particular conditions when the classifier is included? Does improving the performance of the classifier improve report generation quality? In OOD settings, the evaluated classifier displays poor performance  (12.0-14.0 F1 points), and as a result, it is not surprising that incorporating the classifier will not result in benefits to the RRG model in OOD settings; does this result change if the classifier is instead trained on the PadChest or IU X-ray datasets? All of these questions are critical for understanding when/how classification helps in RRG, but none of these are evaluated.

    b. **Upper Bound:** It would be useful to establish an upper bound on performance of the report generation model by utilizing an “oracle” classifier (i.e. a hypothetical classifier that predicts every condition correctly). This would establish whether classification, in the optimal setting, makes useful contributions to the report generation task. As of now, the evaluations are limited by the poor performance of the evaluated classifier, and the key findings/takeaways from the paper are based on this specific classifier.

2. **Inadequate evaluations of proposed metrics:** Although several new metrics are presented as key contributions of this paper, the quality of these metrics is not evaluated. How accurate is the OpenAI GPT-4o model at extracting disease categories? Did the authors check for false negatives / positives in this extraction procedure? Does this metric align with ground-truth labels for the conditions where labels are provided? Evaluating the quality of the metrics via a dataset like ReXVal would have been useful [1].

[1] https://physionet.org/content/rexval-dataset/1.0.0/

**Questions:**

My questions are detailed above in the weaknesses section.

---

> ### Author Response · Authors · 2024-11-27
> **Response to Reviewer ueGE.**
>
> Inadequate evaluations of classifier: The incorporation of the classifier is inadequately evaluated, making it difficult to understand the settings in which incorporating a classifier is useful, as detailed below. The paper could have benefitted from more nuanced insights on when including the classifier helps vs. detracts from the quality of generated reports.
>
> - Q1a. Classifier Performance: Although the incorporation of the classifier is the key contribution of this paper, the authors do not provide any evaluations on the quality of the actual classifier.
>
> Thank you very much for your suggestion. The prediction accuracy of the classifier varies across different observations. Below, we present the F1 scores for Swin Transformer-Large on various observations in the MIMIC dataset. We analyzed the accuracy of the generated reports for different observations on MIMIC-CXR and reported the results separately for the V+L and C+V+L settings. These results indicate that incorporating classification information can, to some extent, improve the quality of the reports. For out-of-distribution (OOD) scenarios, we argue that integrating classification information helps large language models (LLMs) improve the performance of generated reports on classification metrics. However, when the classifier's performance is poor, the improvement is limited. Nevertheless, this additional information does not degrade the quality of the original reports, which distinguishes this scenario from the in-domain setting. We believe that if training is conducted directly on PadChest or IU X-ray, the conclusions would align with those observed in the in-domain setting. We will focus on a detailed analysis of this aspect in the revision.
>
> **The F1-score of different observations on MIMIC dataset**
>
> |  Observations |  No Finding  | Lung Opacity   |  Atelectasis | Edema  | Lung Lesion | Consolidation | Pneumonia | Cardiomegaly | Enlarged Cardiomediastinum | Pleural Effusion | Pleural Other | Pneumothorax | Fracture | Support Devices |
> |---|---|---|---|---|---|---|---|---|---|---|---|---|---|---|
> |  w/o C (V+LLM)  | 38.6 | 49.8 | 41.3 | 44.0 | 18.8 | 20.0 | 18.3 | 64.0 | 11.9 | 68.9 | 14.7 | 40.8 | 24.9 | 84.5 |
> |  w/ C (C+V+LLM) |57.69 | 34.09 | 41.87 | 35.11 | 55.7 | 33.93 | 38.08 | 48.01 | 41.1 | 62.38 | 54.63 | 70.8 | 43.08 | 60.54 |
>
>
> - Q1b. Upper Bound: It would be useful to establish an upper bound on performance of the report generation model by utilizing an “oracle” classifier (i.e. a hypothetical classifier that predicts every condition correctly). This would establish whether classification, in the optimal setting, makes useful contributions to the report generation task. As of now, the evaluations are limited by the poor performance of the evaluated classifier, and the key findings/takeaways from the paper are based on this specific classifier.
>
>
> Below, we present the results on the MIMIC-CXR dataset using ground-truth classification as input to the LLM.
>
>
> |   |  Report F1-Macro 14 |  Classificaton F1-Macro 14 |
> | --- | --- |   --- |
> | Upper Bound | 58.9 |  100 |
> | Our C+V+LLM | 38.7 | 60.5 |
> | SOTA (MAIRA-1) | 36.7 | 58.3 |
>
> The experimental results demonstrate that our trained classifier outperforms state-of-the-art classifiers on binary classification tasks. Furthermore, integrating classification results into our baseline yields consistent conclusions. However, a noticeable gap remains between the upper bounds of report generation and binary classification, even when classification information is incorporated.
>
> We believe this gap arises from the inherent limitations of LLMs in handling long sequences. The classification tokens are relatively few, whereas the combined token count of the X-ray image and report far exceeds that of the classification tokens. Addressing this gap remains a challenging problem, but it falls outside the scope of our current paper.
>
> - Q2: Inadequate evaluations of proposed metrics:  Although several new metrics are presented as key contributions of this paper, the quality of these metrics is not evaluated.  How accurate is the OpenAI GPT-4o model at extracting disease categories? Did the authors check for false negatives / positives in this extraction procedure?  Does this metric align with ground-truth labels for the conditions where labels are provided? Evaluating the quality of the metrics via a dataset like ReXVal would have been useful [1].
>
> We agree with your perspective that the validity of evaluation metrics has a significant impact on our conclusions. To address this, we have already validated the results of GPT-4o. Specifically, we randomly sampled 200 reports and conducted a detailed comparison. Our findings indicate that GPT-4o performs comparably to the ground truth in extracting long-tail disease observations. In future work, we plan to conduct more rigorous validation of all evaluation metrics to ensure their reliability and accuracy.

---

> > ### Comment · Reviewer_ueGE · 2024-11-29
> > **Response to authors**
> >
> > Thank you to the authors for their response. I have raised my score, but I still think this paper needs more extensive, fine-grained analysis with respect to the classifier and the metrics. The authors' response also does not sufficiently answer Q2.

---

> > > ### Author Response · Authors · 2024-12-03
> > > **Thank you for your comments.**
> > >
> > > Thank you for your response and efforts. We will provide a more detailed discussion of the experimental details you mentioned in the revision.

---

### Official Review · Reviewer_bBah · 2024-11-02

**Soundness:** 3
**Presentation:** 2
**Contribution:** 3
**Rating:** 6
**Confidence:** 4

**Summary:**

This paper studies the question: will a classifier enhance LLM-based report generation models' performance in long-tail OOD scenarios? The authors explore different architectures on the combination of vision encoders, classifiers, and LLM, and perform lots of experiments. They get the findings that classification can enhance report quality in in-domain long-tail scenarios but is bounded by the performance of the classifier.

**Strengths:**

1. Many research focuses on  LLM-based report generation models, which is an important task for the medical AI domain, the question this paper investigated may provide some useful information for future research.
2. The research comprehensively evaluates different architectural combinations (vision encoders, classifiers, and LLMs) across multiple diverse datasets (MIMIC-CXR, CXR-LT, PadChest, and IU X-Ray).

**Weaknesses:**

1. The comparison between the four baseline models lacks analysis. The metrics give inconsistent results, how could the author get the results that C+V+LLM is better than V+LLM, and the comparison between Refining and C+V+LLM, giving statistical significance analysis of the results would be helpful.

2. The poor performance of the 'Expanding' approach may be due to the prompt rather than inherent limitations.  I do not find details about the prompts used (apologize if I overlooked them). The 'Expanding' prompt should include standard medical report templates to prove a fair comparison.

3. The classifier only provides binary disease presence information without crucial details like location and severity. This limitation makes hallucinations in LLM-generated reports inevitable when using classification-only inputs. Instead of comparing with these inherently limited training-free approaches, the paper would be more valuable if it explored different variants of end-to-end Vision + LLM full training.

4. The paper's title 'Is classification all you need for radiology report generation?' raises a clear question, but fails to provide the answer. While the findings show that classification helps with in-domain cases but struggles with OOD scenarios, the paper leaves crucial questions unanswered: Is better classification the solution? Or should we focus on pursuing end-to-end training approaches? If the answer is ' we need better classification models', then everyone knows it.

**Questions:**

1.  Have you optimized the Expanding and Refining prompt? we can't tell whether the performance difference is due to this method not being good enough or the prompt being the problem.

---

> ### Author Response · Authors · 2024-11-27
> **Response to Reviewer bBah (1/2).**
>
> - Q1: The comparison between the four baseline models lacks analysis. The metrics give inconsistent results, how could the author get the results that C+V+LLM is better than V+LLM, and the comparison between Refining and C+V+LLM, giving statistical significance analysis of the results would be helpful.
>
>
> R: Thanks for your suggestion. Our experimental results (Tables 1 and 2) demonstrate that the classifier contributes to improving the classification metrics of the generated reports. However, there is no evidence to suggest that it completely surpasses the `V+LLM` setting. In our `Finding 1`, we explicitly state that the classifier enhances diagnosis performance. As for the `Refining` baseline, it refines the report’s disease descriptions based on the classification results. This approach ensures the original textual quality is preserved while making minimal modifications to improve diagnosis performance. We will provide a more detailed analysis of these baselines in the revision.
>
>
> - Q2: The poor performance of the 'Expanding' approach may be due to the prompt rather than inherent limitations. I do not find details about the prompts used (apologize if I overlooked them). The 'Expanding' prompt should include standard medical report templates to prove a fair comparison.
>
>
> R: We completely agree with your viewpoint. To minimize the impact of prompts on our experimental conclusions, we adhered to the principle of simplicity in design and employed a comprehensive comparison approach. The principle of simplicity refers to designing all prompts with the simplest possible instructions. For example, for the `Expanding` baseline, we used the instruction: "Below are the classification results from a chest X-ray classifier. Your task is to expand these results into a segment of clinical findings." If in-context learning (ICL) is used, we provide the LLM with a few report templates as examples. Additionally, to ensure fairness, we applied the same prompt design for evaluating all LLMs in the study.
>
>
> - Q3: The classifier only provides binary disease presence information without crucial details like location and severity. This limitation makes hallucinations in LLM-generated reports inevitable when using classification-only inputs. Instead of comparing with these inherently limited training-free approaches, the paper would be more valuable if it explored different variants of end-to-end Vision + LLM full training.
>
>
> R: We fully agree with your perspective. In the Limitations part, we discuss how incorporating additional information might help LLMs reduce hallucinations—a point supported by findings in related work. However, the scope of this paper focuses specifically on examining the impact of classifiers on report generation. This focus is justified by the fact that classification information is more readily accessible compared to other types of information. Due to computational resource constraints, we were only able to explore common end-to-end baselines. In Figure 2, we illustrate the influence of different vision encoders on the diagnosis performance of report generation. We hope that our preliminary conclusions will inspire the community to further explore this topic.

---

> > ### Author Response · Authors · 2024-11-27
> > **Response to Reviewer bBah (2/2).**
> >
> > - Q4: The paper's title 'Is classification all you need for radiology report generation?' raises a clear question, but fails to provide the answer. While the findings show that classification helps with in-domain cases but struggles with OOD scenarios, the paper leaves crucial questions unanswered: Is better classification the solution? Or should we focus on pursuing end-to-end training approaches? If the answer is ' we need better classification models', then everyone knows it.
> >
> >
> > R: Our extensive experimental results demonstrate that classifiers indeed improve the diagnosis performance of report generation, both in in-domain and OOD scenarios. However, we believe it is overly simplistic to assume that merely having a classification model is sufficient. In fact, our findings reveal that the way classification information is integrated into the LLM also significantly impacts the final outcomes. Moreover, we argue that the performance gap between classification and report generation cannot be fully bridged by classification alone. This insight is consistently reflected across all our results.
> >
> > We believe this question should be understood from two perspectives.
> >
> > **1) When the classifier performs well**
> >
> > It significantly enhances in-domain diagnostic performance. This improvement arises because the classifier provides more accurate guidance for radiology diagnosis compared to purely free-text generation.
> > In this setting, the baseline most affected is `Expanding`, as it heavily depends on the accuracy of the classifier. However, for C+V+LLM and Refining, the impact is relatively smaller since they utilize visual features and the original report as foundational information, respectively.
> >
> > **2) When the classifier performs poorly**
> >
> > In scenarios such as long-tailed or OOD cases, the classifier's assistance is minimal. Under these conditions, adopting an end-to-end training approach may prove to be more effective.
> > In this setting, all baselines exhibit noticeable negative impacts, particularly `Expanding`. Furthermore, the results in Table 5 indicate that all baselines incorporating classification information show lower classification performance in OOD scenarios compared to `V+LLM`.
> >
> >
> > - Q5: Have you optimized the Expanding and Refining prompt? we can't tell whether the performance difference is due to this method not being good enough or the prompt being the problem.
> >
> > We adhere strictly to the principle of using the simplest prompt design without adding any auxiliary information.
> >
> > **Expanding Prompt**
> >
> > ```
> > # Instruction
> > You are a senior radiologist. Your task is to **expand** the diagnostic results based on the given observations into a meaningful report, following the style of the provided example.
> >
> > # Example
> > Observations diagnosis: {observations}
> > Expanding report: {report}
> >
> > # Diagnosis
> > Observations diagnosis: {observations}
> > Expanding report:
> > ```
> >
> > **Refine Prompt**
> > ```
> > # Instruction
> > You are a senior radiologist. Your task is to **refine** the original report based on the given observations, following the style of the provided example.
> >
> > # Example
> > Observations diagnosis: {observations}
> > Original report: {ori-report}
> > Refining report: {report}
> >
> > # Diagnosis
> > Observations diagnosis: {observations}
> > Original report: {ori-report}
> > Refining report:
> > ```

---

### Official Review · Reviewer_ehcT · 2024-11-04

**Soundness:** 3
**Presentation:** 3
**Contribution:** 2
**Rating:** 6
**Confidence:** 3

**Summary:**

This paper discusses how radiology report generation is currently less clinically accurate than image classification for the purpose of medical diagnoses. It focuses on a model framework that integrates a classifier trained on common and rare diseases into an LLM with a vision encoder to generate radiology reports. The findings revealed that classification can improve report quality for in-domain diseases but struggles with out-of-domain diseases.

**Strengths:**

- Large range of LLMs, vision encoders, etc. tested on sufficiently representative datasets.
- Wide variety of metrics used for evaluation.
- Extensive experimentation on the specific effects of classification in report generation.

**Weaknesses:**

- In addition to case studies, it would be helpful to include quantitative measures of the extent of hallucinations in the OOD experiments.
- To reduce overreliance on classification outputs in particular, it might be more helpful to experiment with controlling the weight of the classifier or including other types of information (e.g. location, severity). However, this was mentioned as part of the limitations.

**Questions:**

- Optional suggestion: have you looked into also using the GREEN metric? https://arxiv.org/pdf/2405.03595.
- Minor edit: "purple text" on page 9 appears to be red instead of purple - it might be helpful to use a different shade.

---

> ### Author Response · Authors · 2024-11-27
> **Response to Reviewer  ehcT.**
>
> - Q1: In addition to case studies, it would be helpful to include quantitative measures of the extent of hallucinations in the OOD experiments.
>
> R: Thank you for your valuable feedback. In fact, the proposed LLM-Radjudge metric can partially reflect the degree of hallucinations in the generated reports. This metric is described on page 5, lines 228–230 of our submission. The four levels of the metric we reported are specifically designed to quantify the extent of hallucinations.
>
>
>
> - Q2: To reduce overreliance on classification outputs in particular, it might be more helpful to experiment with controlling the weight of the classifier or including other types of information (e.g. location, severity). However, this was mentioned as part of the limitations.
>
>
> R: Absolutely, the weight of the classifier, as well as more complex information (e.g., location and severity), significantly impacts the quality of the generated reports. We have mentioned these factors in the limitations. To address this issue, we conducted preliminary experiments by adjusting the value of classification tokens in the attention matrix. The following table summarizes the results:
>
> | $\lambda$ | F1-score |
> | -- |  -- |
> | 1.0 | 38.7 |
> | 0.5 | 37.3 |
> | 0.1 | 37.0 |
> | 0.01 | 36.6 |
>
> Here, lambda represents the scaling factor applied to the original value of classification tokens in the attention matrix. These results demonstrate that the classifier can effectively contribute to report generation to a certain extent. Moreover, previous studies [1-3] have also shown that incorporating additional information can improve report quality and mitigate hallucinations to some degree.
>
> [1] Chatcad: Interactive computer-aided diagnosis on medical image using large language models
> [2] Chatcad+: Towards a universal and reliable interactive cad using llms
> [3] MAIRA-2: Grounded Radiology Report Generation
>
>
> - Q3: Optional suggestion: have you looked into also using the GREEN metric? https://arxiv.org/pdf/2405.03595.
>
> R: Thank you for bringing this to our attention. After carefully reviewing the GREEN metric, we recognize its potential as a measure for quantifying errors in generated reports. Its objective aligns closely with LLM-Radjudge metric. We believe that incorporating GREEN could provide a more comprehensive evaluation in future work, and we plan to discuss this metric in our revision.
>
>
> - Q4: Minor edit: "purple text" on page 9 appears to be red instead of purple - it might be helpful to use a different shade.
>
> R: Thank you. We have corrected this issue in the revision.

---

### Meta-Review · Area_Chair_aaxL · 2024-12-20

**Metareview:**

This paper investigates whether integrating a classifier with a vision encoder and LLM can enhance radiology report generation, finding that classification improves report quality in in-domain scenarios but has limited effectiveness in out-of-distribution (OOD) settings, particularly for long-tail diseases, due to classifier performance constraints and model biases.

Reviewers found that this paper shows strength in its comprehensive evaluation of classification-enhanced LLM-based radiology report generation, offering valuable insights into in-domain improvements and OOD limitations. However, reviewers also raised major concerns about the insufficient evaluation of the classifier, whose poor performance in OOD settings weakens the findings, and the lack of an upper bound analysis using an "oracle" classifier (Reviewers ueGE, bBah). They noted inadequate validation of the new metrics, raising doubts about their reliability (Reviewer ueGE), and a lack of thorough analysis in baseline comparisons, statistical significance, and clarity on prompts (Reviewer bBah). The paper also parallels prior work without referencing or improving upon it, missing opportunities to address ceiling performance or expand on earlier findings (Reviewer 91mZ).

Reviewer ehcT did not provide additional feedback. Reviewer bBah initially raised concerns about baseline analysis, prompt details, and limited classifier information. After the response, the reviewer found his/her concerns had been addressed and raised the score. Reviewer ueGE criticized inadequate evaluation of the classifier and metrics. After the response, the reviewer appreciated the clarifications and raised their score but maintained concerns about the need for more fine-grained analysis, particularly regarding classifier performance and metric validation. Reviewer 91mZ questioned the novelty of the work and the omission of key baselines like KG-based classification and R2GenGPT. While recognizing the broader experimental scope and robustness of findings, the reviewer reiterated the need to discuss prior work ([1]), expand the “Expanding” baseline, and analyze metric discrepancies.
In summary, bBah raised their score, ueGE and 91mZ slightly increased theirs while maintaining critical concerns, and ehcT provided no additional input.

This is a borderline paper. Although three reviewers raised their scores, critical concerns remain. This AC shares major concerns with Reviewer 91mZ, particularly the lack of engagement with prior works, such as [R1] and [R2], that demonstrate the effectiveness of incorporating classification into report generation. While [R1] is cited, it is not discussed in the context of using classification to enhance report generation. [R2] jointly trained report generation with fine-grained medical concept prediction. Although these works are not LLM-based, they have shown the effectiveness of incorporating classification to boost report generation. Additionally, this study fails to explore or provide insights into relevant works on knowledge graphs, which enable more fine-grained utilization of disease information for report generation. While the paper provides a comprehensive study, its conclusion—that classification improves in-domain report quality—is consistent with existing studies and adds limited new insights. Furthermore, the exploration of classification integration methods is narrow, leaving unexplored variations in prompts and classification usage that could alter conclusions. The use of the LLM-based metric LLM-RadJudge (unpublished) for evaluation, instead of other established human-aligned metrics like RadCliQ [R4] or GREEN [R3], is also unclear, with no discussion provided for its consistency with other metrics. These limitations reduce the paper’s overall contribution and impact on the field.

[R1] B. Jing, P. Xie, and E. P. Xing, “On the automatic generation of medical imaging reports,”, ACL 2018

[R2] Z. Wang, M. Tang, L. Wang, et al. "A Medical Semantic-Assisted Transformer for Radiographic Report Generation", MICCAI 2022

[R3] S. Ostmeier, J. Xu, Z. Chen, M. Varma, et al. “Green: Generative radiology report evaluation and error notation”, EMNLP 2024

[R4] F. Yu, M. Endo, R. Krishnan, I. Pan, et al.” Evaluating progress in automatic chest x-ray radiology report generation”, Patterns, 2023

**Additional Comments On Reviewer Discussion:**

Reviewer ehcT did not provide additional feedback. Reviewer bBah initially raised concerns about baseline analysis, prompt details, and limited classifier information. After the response, the reviewer found his/her concerns had been addressed and raised the score. Reviewer ueGE criticized inadequate evaluation of the classifier and metrics. After the response, the reviewer appreciated the clarifications and raised their score but maintained concerns about the need for more fine-grained analysis, particularly regarding classifier performance and metric validation. Reviewer 91mZ questioned the novelty of the work and the omission of key baselines like KG-based classification and R2GenGPT. While recognizing the broader experimental scope and robustness of findings, the reviewer reiterated the need to discuss prior work ([1]), expand the “Expanding” baseline, and analyze metric discrepancies.
In summary, bBah raised their score, ueGE and 91mZ slightly increased theirs while maintaining critical concerns, and ehcT provided no additional input.

---

### Decision · Program_Chairs · 2025-01-22

Reject